# Aggregated Tau-PHF6 (VQIVYK) Potentiates NLRP3 Inflammasome Expression and Autophagy in Human Microglial Cells

**DOI:** 10.3390/cells10071652

**Published:** 2021-06-30

**Authors:** Chinmaya Panda, Clara Voelz, Pardes Habib, Christian Mevissen, Thomas Pufe, Cordian Beyer, Sharad Gupta, Alexander Slowik

**Affiliations:** 1Biological Engineering, Indian Institute of Technology Gandhinagar, Palaj, Gandhinagar 382355, India; chinmayapanda9732@gmail.com (C.P.); sharad@iitgn.ac.in (S.G.); 2Institute of Neuroanatomy, Medical Faculty, RWTH Aachen University, 52074 Aachen, Germany; cvoelz@ukaachen.de (C.V.); cbeyer@ukaachen.de (C.B.); 3Department of Neurology, Medical Faculty, RWTH Aachen University, 52074 Aachen, Germany; phabib@ukaachen.de; 4Institut für Organische Chemie, RWTH Aachen University, 52074 Aachen, Germany; christian.mevissen@rwth-aachen.de; 5Department of Anatomy and Cell Biology, Medical Faculty, RWTH Aachen University, 52074 Aachen, Germany; tpufe@ukaachen.de

**Keywords:** Alzheimer’s disease, PHF6, tau, microglia, HMC3, autophagy, NLRP3

## Abstract

Intra-neuronal misfolding of monomeric tau protein to toxic β-sheet rich neurofibrillary tangles is a hallmark of Alzheimer’s disease (AD). Tau pathology correlates not only with progressive dementia but also with microglia-mediated inflammation in AD. Amyloid-beta (Aβ), another pathogenic peptide involved in AD, has been shown to activate NLRP3 inflammasome (NOD-like receptor family, pyrin domain containing 3), triggering the secretion of proinflammatory interleukin-1β (IL1β) and interleukin-18 (*IL18*). However, the effect of tau protein on microglia concerning inflammasome activation, microglial polarization, and autophagy is poorly understood. In this study, human microglial cells (HMC3) were stimulated with the unaggregated and aggregated forms of the tau-derived PHF6 peptide (VQIVYK). Modulation of NLRP3 inflammasome was examined by qRT-PCR, immunocytochemistry, and Western blot. We demonstrate that fibrillar aggregates of VQIVYK upregulated the NLRP3 expression at both mRNA and protein levels in a dose- and time-dependent manner, leading to increased expression of IL1β and *IL18* in HMC3 cells. Aggregated PHF6-peptide also activated other related inflammation and microglial polarization markers. Furthermore, we also report a time-dependent effect of the aggregated PHF6 on *BECN1* (Beclin-1) expression and autophagy. Overall, the PHF6 model system-based study may help to better understand the complex interconnections between Alzheimer’s PHF6 peptide aggregation and microglial inflammation, polarization, and autophagy.

## 1. Introduction

Alzheimer’s disease (AD), the primary cause of dementia, is currently affecting over 50 million patients worldwide [1]. Currently, there are no available disease-modifying therapies for AD. Misfolded protein deposits such as extracellular beta-amyloid (Aβ) plaques and intra-neuronal tau neurofibrillary tangles (NFTs) are often charged with exacerbating the AD pathology [2]. The severity of AD can be correlated with the deposition of NFTs [3]. Besides the stabilization of the cytoskeleton by promoting tubulin monomeric assembly, the tau protein also regulates the cargo transport along the microtubules [4,5]. Abnormal post-translational modifications promote the conversion of unfolded soluble monomeric proteins into insoluble neurotoxic oligomers, so-called β-sheet rich paired helical filaments (PHFs), and NFTs [6]. These protein aggregates instigate neuronal inflammation, eventually aggravating the disease conditions.

The tau core hexapeptide fragment ^306^VQIVYK^311^ (PHF6) is a self-assembly sequence capable of initiating the full-length tau protein aggregation and is mapped to the third microtubule-binding repeat region of the tau protein [7,8]. Cryo-EM structure of full-length tau protein detailed the formation of the N-terminal cross β-sheet structure mediated by the PHF6 fragment [9]. It also highlighted the role of hexapeptide in modulating the phosphorylation patterns and aggregation propensity. PHF6 is extensively used as a model system to understand the aggregation pattern owing to its ability to form β-sheet rich conformation similar to full-length tau aggregates [8,10,11,12].

Apart from proteotoxicity, neuroinflammation triggered by microglia and astrocytes significantly affects neurodegeneration and the progression of AD [13,14]. NOD-like receptor family, pyrin domain-containing 3 (NLRP3), is a vital component of the innate immune system and is found to be activated in AD patients’ brains [15]. In the presence of danger signals, it forms the NLRP3 inflammasome complex in association with ASC (apoptosis-associated speck-like protein containing a CARD) and caspase-1. Subsequently, catalysis of pro-caspase-1 helps to release proinflammatory cytokines, viz., IL1β, and *IL18* [16,17,18]. Activated NLRP3 and reactive gliosis are general characteristics of tauopathies like AD [19].

Age-related protein deposits like Aβ activate astroglial cells via different cell surface receptors such as toll-like receptors *TLR2* or *TLR4* [20], resulting in elevated levels of cleaved caspase-1 (CASP1) and released cytokines [21]. The role of activated microglia in amyloid plaque clearance through phagocytosis has also been reported elsewhere [22,23]. However, limited literature is available regarding full-length tau (or PHF6)-mediated microglial activation in AD. The role of microglia in the uptake, degradation, and release of tau seeds has been demonstrated in transgenic tau mice [24,25]. Apart from tau spreading, microglia might also aid in the aggregation of tau proteins [26]. Recently, aggregated full-length tau protein was linked with activated NLRP3 inflammasome [27].

In a healthy human brain, autophagy processes support the clearance of unnecessary or toxic cell particles to maintain a healthy environment for brain cells. In unspecific autophagy, cell particles are enclosed by an autophagosome, which itself is uptaken by a lysosome to finally build the autophagolysosome. In there, cell particles are digested and recycled for later use [28]. During this process, several proteins play an important role, and their dysregulation could point to an overall dysregulation of autophagy. Beclin-1 is an active component for the formation of autophagy vesicles and is found to be associated with receptor-mediated phagocytosis in microglia. Further, Beclin-1 seems to modulate NLRP3-mediated neuroinflammation [29,30]. Microtubule-associated light chain 3 (LC3), known to be very specific in autophagy processes, plays a role in recruiting the autophagosomes to their targets. LC3 is recognized by specific receptors, e.g., Sequestosome-1 (SQSTM-1, also known as p62) [31]. Over time in an aging brain, factors that cooperate to perform functioning autophagy are expressed to a lesser extent. This could contribute to the late onset of neurodegenerative diseases in aging brains [32]. In 2005, a research group of Nixon et al. found autophagic vacuoles were accumulated in the brains of AD patients [33]. Since then, dysfunction of autophagy in AD has been described, though it is unclear if it is a cause or a consequence of the disease. Additionally, the relationship between AD and autophagy seems to vary in a time dependent manner [34]. The clearance of misfolded tau protein through the chaperon-mediated autophagy (CMA) pathway has also been reported elsewhere [35,36]. Moreover, inflammasome activation has also been shown in other tauopathies, such as frontotemporal dementia [37].

The clearance of tau deposits by microglial cells is a multi-faceted mechanism involving autophagy, phagocytosis, inflammasome activation, and proinflammatory cytokine release. The influence of full tau in AD has been studied more intensely and its role in microglia and NLRP3 activation has been described previously. The role of tau peptides on microglial cells is also relatively unknown and is largely unexplored. Considering the significance of PHF6 in tau aggregation and the spreading of tau seeds, we want to know if PHF6 is also involved/part of the activation processes. Therefore, in the present study we used the PHF6 peptide as a model system to investigate the role of aggregation on microglial polarization, inflammasome activation, and autophagy in the human microglial cell line HMC3.

## 2. Material and Methods

### 2.1. Cell Culture

The human microglial clone 3 (HMC3) cell line was purchased from ATCC^®^ (Product Number: CRL-3304™). The HMC3 cell line is derived from human primary microglial cells and is widely considered to be a unique experimental model for glial experiments [38]. Cells were grown in a humidified setting in an incubator at 37 °C with a 5% CO_2_ supply. The culture medium Gibco™ Dulbecco’s modified Eagle medium (DMEM) was supplemented with 10% fetal bovine serum (FBS) and 0.5% penicillin-streptomycin (P/S) (all from Life Technologies, Waltham, MA, USA). For experiments, 25,000 cells/cm^2^ were seeded on poly-L-ornithine (PLO) coated experimental plates in phenol-red free Gibco^TM^ Roswell Park Memorial Institute (RPMI) 1640 medium (Thermo Fisher Scientific, Waltham, MA, USA) containing 5% FBS and 0.5% P/S. The stimulation of cells with aggregated or unaggregated PHF6 was performed 24 h after seeding (Figure 1).

### 2.2. Aggregation Assay

Tau-PHF6 (Ac-VQIVYK-NH_2_) was purchased from Bachem (Bubendorf, Switzerland). One mg of the peptide was dissolved in 1 mL of ultrapure water. The stock solutions of 55 μM Heparin and 3.14 mM Thioflavin T dye (ThT) (both from Sigma–Aldrich, Darmstadt, Germany) were also prepared in ultrapure water. For the aggregation experiment, as detailed by KrishnaKumar et al. 2018, stock solutions of peptide, heparin, and ThT were diluted in 20 mM MOPS buffer (pH 7.2) so that the final working concentrations in a 96-well Corning^®^ dark flat bottom plate were 50 μM, 10 μM, and 100 μM, respectively [7]. For each well, 100 μL of aggregation mixture was used. Control wells were set up, having water in place of peptide or with the buffer only. Kinetic fluorescent measurement was performed in quadruplicate every 20 s at λ_ex_ (excitation) = 440 nm and λ_em_ (emission) = 485 nm using the Infinite^®^ M200 plate reader (Tecan, Zurich, Switzerland) for over an hour. For cell culture experiments, the aggregation mixture contained peptide and heparin, but no ThT because it could be toxic to the cells. The aggregation mixture was later diluted with the cell culture medium to match the final required concentration on cells. Apart from the kinetic measurement assay, a turbidity assay was also performed at 350 nm (without ThT) using the Tecan plate reader, which helps to observe the difficulty of the light to pass through the aggregate formation over time.

An appropriate number of cells were seeded on 24-well plates in RPMI 1640 media for 24 h before stimulation with the appropriate concentration of aggregated and unaggregated PHF6. After 6 and 24 h stimulation, protein isolation for WB, cell viability assay, immunocytochemistry, RNA isolation and qRT-PCR, and autophagy assay were performed.

### 2.3. Microscopic Analysis of Aggregates

After the ThT assay, fibrils were visualized on a glass-slide by Leica DM6000 B (Leica Microsystems, Wetzlar, Germany) microscope using the green channel at a 63X (oil immersion) objective magnification.

### 2.4. Circular Dichroism (CD) Spectroscopy

CD spectra were measured on an AVIV 62D CD spectrometer. All measurements were performed using a 1 mm glass cuvette. The concentration of the assembled tau-derived PHF6 probe was adjusted to 50 μM. Wavelength-dependent circular dichroism of the probe was recorded in the range of 190–250 nm and plotted against the applied wavelength λ. Further, using the webserver BeStSel [39], a secondary structure prediction was performed from the CD spectroscopy data.

### 2.5. Dose and Time Dependency

To explain the short term and long-term effects of different concentrations of the tau-PHF6 peptide (aggregated and unaggregated) on human microglial HMC3 cells, a dose-dependent study was carried out with increasing concentrations (1, 5, 10, and 20 μM) of the peptide at two different time points (6 and 24 h).

### 2.6. Cell Viability

#### 2.6.1. CTB Assay

The metabolic activity of the cells was assessed using the Cell Titer-Blue^®^ Cell Viability Assay (Promega, Madison, WI, USA), following the manufacturer’s protocol. Briefly, microglial cells were seeded on 96-well plates 24 h before stimulation with tau peptides. After 6 and 24 h tau stimulation, the assay reagent was added to the wells, and after 3 h, due to the reduction of resazurin to resorufin, a color change was observed. Fluorescent measurement was then performed at λ_ex_ = 560 nm and λ_em_ = 590 nm using the Infinite^®^ M200 plate reader.

#### 2.6.2. LDH Release Assay

Cell viability was assessed by the CytoTox 96^®^ Non-Radioactive Cytotoxicity Assay (Promega, Madison, WI, USA), which measures the release of lactate dehydrogenase (LDH) into the growth medium, indicating leakage of the outer cell membrane. Following the manufacturer’s protocol, microglial cells were seeded on 24-well plates 24 h before stimulation. After 6 and 24 h of peptide stimulation, 50 μL of the supernatant was transferred to a 96-well plate, and then 50 μL of the assay reagent was added to each well. Apart from the samples, media-only control and a positive control containing lysed cells were also used. The absorbance values were recorded at λ = 490 nm using the Tecan Infinite^®^ M200 plate reader. All values were represented as the percentage of the lysis control.

### 2.7. Gene Expression Analysis

After 6 and 24 h of stimulation, a phenol-chloroform-based extraction method was used for the RNA isolation [40]. The growth medium was removed and replaced with the peqGOLD TriFast™ (Peqlab, Darmstadt, Germany) reagent according to the manufacturer’s protocol. NanoDrop™ 1000 (Thermo Fisher Scientific, Waltham, MA, USA) reported the RNA purity and concentration. RNA samples were regarded as pure if the 260/280 ratio indicated 2.0 ± 0.1. For the reverse transcription, 1 μg/mL of total RNA was used to produce cDNA using the Moloney murine leukemia virus (M-MLV) reverse transcription kit and random primers (both from Invitrogen™, Waltham, MA, USA). Semi-quantitative polymerase chain reaction (PCR) was performed with reference housekeeping gene primers *B2M* (β2-microglobulin) and HPRT (hypoxanthine phosphoribosyltransferase 1) (Table 1). Mastercycler gradient S (Eppendorf, Hamburg, Germany) was used for the reaction. The PCR run protocol was as follows: 95 °C for 3 min, at 95 °C for 40 s (denaturation), at the annealing temperature of the primer (Table 2) for 40 s, at 72 °C for 45 s, at 72 °C for 45 s (extension), followed by 72 °C for 2 min and final hold at 4 °C. The steps from denaturation up to extension were repeated 40 times. Nucleic acid bands were separated by gel electrophoresis (constant voltage of 160 V for 20 min) using 2% agarose gels containing Midori Green Advance DNA staining dye (Nippon Genetics, Duren, Germany). Gels were finally visualized in Vilber E-box VX2 (Peqlab, Darmstadt, Germany).

For quantitative real-time PCR (qRT-PCR), the samples were diluted 10-fold with ultrapure water. Appropriate amounts of AceQ qPCR SYBR^®^ Green (CliniSciences, Nanterre, France), primers (Table 1), and ultrapure water were used for the master mixture. CFX Connect™ Real-Time PCR Detection System (Bio-Rad, Hercules, CA, USA) was used with the following configuration: at 95 °C for 10 min (enzyme activation), at 95 °C for 15 s (denaturation), at the annealing temperature of the primers (Table 1) for 30 s, at 72 °C for 30 s (amplification), followed by a melting curve analysis. The cycle of denaturation up to amplification was repeated 40 times. All target gene expressions were normalized to the expression levels of the housekeeping genes (*B2M* and *HPRT*). Apart from the analysis using CFX Maestro software (Bio-Rad, Hercules, CA, USA), gel electrophoresis was also performed to evaluate the specificity of amplified products.

### 2.8. Western Blot

After stimulation of microglial HMC3 cells for 6 and 24 h, protein isolation was performed using the radioimmunoprecipitation assay buffer (RIPA), supplemented with protease inhibitor cOmplete mini™ (Sigma–Aldrich, Darmstadt, Germany). For measuring the protein concentration, Pierce™ BCA Protein Assay Kit (Thermo Fisher Scientific, Waltham, MA, USA) was used with absorbance recorded at λ = 562 nm using the Tecan Infinite^®^ M200 plate reader. After heating the samples at 95 °C for 8 min, they were loaded (20 μg protein per lane) onto the gels, and electrophoresis was performed at 80 V for 15 min followed by 140 V for 1 h. Blotting of the gel onto methanol-activated polyvinylidene difluoride (PVDF) membranes (Trans-Blot^®^ Turbo™ RTA Mini PVDF Transfer Kit, Bio-Rad, Hercules, CA, USA) was performed using the Trans-Blot^®^ Turbo™ Transfer System (Bio-Rad, Hercules, CA, USA) at 25 V for 50 min. Ponceau S was used to verify the success of the blotting. After re-activation of the membrane with methanol and subsequent blocking for 1 h with 5% milk in TBS, membranes were incubated with the primary antibody overnight at 4 °C in the blocking solution. On the next day, after washing, the secondary antibody was added to visualize the immunoreactive signals using the Vilber Fusion Solo detection system (Peqlab, Darmstadt, Germany). Table 2 lists the antibodies used for Western blot. For chemiluminescence detection of bands, the Westar Supernova (Cyanagen, Bologna, Italy) substrate reagent was used. Using ImageJ software (NIH, Bethesda, MD, USA), densitometric analysis of bands was performed and later normalized to the β-actin housekeeper protein control bands.

### 2.9. Immunocytochemistry

Microglial HMC3 cells were seeded on PLO-coated coverslips, and after stimulation and proper incubation time, cells were 3.7% PFA-fixed. Then, cells were washed with phosphate-buffered saline (PBS) and permeabilized with Triton-X. After blocking, incubation with the primary antibody was done overnight at 4 °C. After washing, a fluorescent-labeled secondary antibody was added for 2 h. The nuclei were counterstained with Hoechst 33342, Trihydrochloride, Trihydrate (Invitrogen™, Waltham, MA, USA). Table 3 lists all antibodies used for ICC. The Leica DM6000 B microscope was used to capture fluorescence images, with identical microscope settings maintained for each experiment.

Fluorescence intensity was evaluated by FIJI software [41]. The mean grey value was determined per sample and the background was subtracted. Afterwards, relative quantity to the control group was calculated. A total of four pictures were used per group and experiment. The fluorescence intensity is a linear grade and can be used for determination of protein levels when the same exposure settings are used for acquisition [42].

### 2.10. Caspase-1 Activity Kit

The CASP1 activity for the determination of inflammasome activation was determined by using the Caspase-Glo^®^ 1 Inflammasome Assay (Promega, Madison, WI, USA), following the manufacture’s protocol. In brief, half of the supernatant was removed and replaced with the 1:2 dilution of the ready-to-use Glo1 reagent and incubated for 60 min at room temperature. For NLRP3 inhibition, the cells were pre-incubated for 1 h with the NLRP3-specific inhibitor MCC950 (1 µM; AdipoGen Life Science, Liestal, Switzerland). Finally, luminescence signals were read out with the Tecan Infinite^®^ M200 plate reader.

### 2.11. ELISA

After recovering the supernatants from the culture plates, protein concentration was measured by the Pierce™ BCA Protein Assay Kit (Thermo Fisher Scientific, Waltham, MA, USA) with absorbance recorded at λ = 562 nm using the Tecan Infinite^®^ M200 plate reader. Human IL1β/IL1F2 DuoSet ELISA (R&D Systems, Minneapolis, MN, USA) was performed as per the manufacturer’s protocol. Specialized ELISA 96-well plate was incubated with the captured antibody overnight, followed by 1 h of blocking and then the addition of standard/samples overnight. Next, a detection antibody was added for 2 h at room temperature. Then, Streptavidin-HRP conjugate was added for 20 min, followed by the addition of substrate solution for 20 min. Nigericin (20 µM, Sigma–Aldrich, Darmstadt, Germany), LPS (1 µg/mL, Sigma–Aldrich, Darmstadt, Germany), or the combination of both reagents, were used as positive controls. The Tecan Infinite^®^ M200 plate reader was then used to measure the color development at 405 nm.

### 2.12. Detection of Autophagosome Formation

Because autophagy association has previously been reported in the case of Aβ [29], the formation of autophagosomes due to tau-PHF6 stimulation was tracked using the CYTO-ID^®^ autophagy detection kit 2.0 (Enzo Life Sciences, Farmingdale, NY, USA). A mixture of rapamycin (0.5 μM) and chloroquine (10 μM) was used as a positive control, as specified by the manufacturer. For microscopic analysis, Hoechst and CYTO-ID^®^ detection reagent 2 (1:500 final dilution) were directly added to the cells growing in 24-well plates. Cells were imaged using the Leica DM6000 B microscope with a 63X oil immersion objective. For the plate reader measurement, the detection reagent 2 (1:1000 final dilution) and Hoechst were added to cells growing in 96-well plates, and a fluorescence measurement was recorded at λ_ex_ = 480 nm and λ_em_ = 530 nm using the M200 Tecan plate reader.

### 2.13. Analysis

Experiments were conducted in triplicates (gene expression) or quadruplets (all other experiments), and data are represented as the arithmetic means ± SEM. In prior experiments, the needed *n* values were calculated using G*Power [43] according to the first gene expression data with the following settings: *f* = 1.6; α = 0.05; β = 0.9; 3 groups. G*Power calculated, for the gene expression analysis, that 3 *n* per group were needed. Because we used gene expression data for the calculation for the further experiment, we added one extra *n* for the other experiments, due to lower effects being expected. Before analysis, using the Shapiro–Wilk normality test, a Gaussian distribution check of residual data points was performed. Further, using the Bartlett test, an equal variance was evaluated. If one of the tests returns significance, data transformation was performed using the Box-Cox method, and reevaluation of the tests was done for the newly generated transformed data points. Finally, a one-way or two-way (only for grouped analyses) ANOVA, followed by Tukey’s post-hoc test for inter-group differences, was performed. Statistical significance was set at 95% confidence interval with *p* values set as */# ≤ 0.05, **/## ≤ 0.01 ***/### ≤ 0.001.

## 3. Results

### 3.1. PHF6 Rapidly Forms Aggregates

When incubated in the presence of heparin, PHF6 peptide solution containing ThT reported a sharp increase in fluorescence in 10–15 min, attaining a sigmoid curve with a distinct lag phase and a plateau (Figure A1a). This effect strongly suggests the presence of high amyloidogenic aggregated species in the mixture [44,45]. The turbidity assay also supported the aggregated species formation (Figure A1b). CD spectroscopy data depicted a strong positive peak at 196 nm and a negative curve between 210–215 nm, which are the characteristics of β-sheet structures [46,47]. Predictions from the BeStSel server also indicated a high percentage of β-sheet along with random coil. Finally, for the cell experiments, two different forms of PHF6 were used, one being the aggregated sample with β-sheet structures (denoted as aPHF6) and the other being unaggregated monomeric PHF6 (denoted as PHF6).

### 3.2. aPHF6 Affects the Metabolic Activity of Microglia in A Time- and Concentration-Dependent Manner

The HMC3 cells were stimulated with aPHF6 for 6 and 24 h to identify the short- and long-term effects on cell viability. Cell toxicity measured by the LDH release after 6 h of stimulation with 20 μM PHF6 reached 10% compared to the lysed cells (positive control). After 24 h of exposure, the LDH release was around 20% of the positive control (Figure 2a). There was no significant dose-dependent cytotoxic effect of aPHF6 or PHF6 on microglial cells across the used concentrations of 1–20 μM (Figure A2a). As indicated with the CTB assay, after 6 h of stimulation with 20 μM aPHF6 or PHF6, the metabolic activity almost doubled compared to the untreated control (Figure 2b), whereas with lesser concentrations of aPHF6 and PHF6 (1–10 μM), the metabolic activity after 6 h also increased but remained around 1.5 times that of the control (Figure A2b). After 24 h of stimulation, however, the metabolic rate decreased to the control levels across all different groups with no dose-dependency being observed.

### 3.3. aPHF6 Induced NLRP3 mRNA and Protein Upregulation

In the first step, the expression of NLRP3 after aPHF6 stimulation in HMC3 cells was evaluated. For this, a trial was done with four different concentrations of the aPHF6 (1, 5, 10, and 20 μM) and two different concentrations of PHF6 (10 and 20 μM) across two different time points (6 and 24 h). A sharp increase in the mRNA level of NLRP3 was observed with 20 μM aPHF6 stimulation at 6 h compared to the control, which eventually returned to the basal level after 24 h (Figure A3). With the lesser concentrations of aPHF6 or PHF6 (1–10 μM), the gene expressions were comparable to the control levels.

Hence, 20 μM aPHF6 and 20 μM PHF6 samples were finally selected for further evaluation. With the aPHF6 stimulation, a seven-fold increase in the NLRP3 gene expression was observed after 6 h compared to the control (Figure 3a). In contrast, after 24 h, the gene expression level indicated only a 1.5-time increase (Figure 3d). For the 20 μM PHF6, however, no such increase was observed. Densitometric analysis of Western blots also depicted an almost two-fold increase in the protein expression levels after 6 h of aPHF6 stimulation in HMC3 cells compared to the control group (Figure 3b), whereas after 24 h, no shifts were detected (Figure 3e). With the 20 μM PHF6, however, no significant increase in protein expression was observed at both time points. Immunofluorescence staining of cells was performed using Iba1, NLRP3, and Hoechst (nuclear staining). Around a 1.5-fold increase in NLRP3 fluorescence intensity after 6 h of 20 μM aPHF6 stimulation was observed, as quantified by ImageJ (Figure 3c). After 24 h, however, NLRP3 intensity was found comparable to the control levels across all different groups, with no significant increase in protein expression being observed (Figure 3f).

### 3.4. aPHF6 Induced Upregulation of ASC (PYCARD) and M1 Markers

The adapter ASC (*PYCARD*) is an active component of the active NLRP3 inflammasome. Similar to *NLRP3*, a time-dependent increase in the expression levels of mRNA and protein was also witnessed with *PYCARD*/ASC in aPHF6-induced HMC3 cells. After 6 h of aPHF6 stimulation, the gene expression increased by 15-fold compared to the control (Figure 4a). However, after 24 h of stimulation, the gene expression significantly abated to the basal levels of the untreated control. Western blot analysis also depicted a similar trend of about a 2.5-fold increase in the ASC protein expression after 6 h of aPHF6 stimulation (Figure 4b), whereas, after 24 h, the increase was only two-fold with respect to the control. With the 20 μM PHF6, however, no significant increase was observed at both time points. To evaluate the effect of aPHF6 on microglial inflammation, gene expression levels of several inflammation markers such as *NLRP1*, *IL12A*, *IL18*, and *TLR4* were also checked (Figure A4). Compared to the control, *NLRP1* and *IL12A* registered about a 5- and 15-fold increase in the gene expression, respectively, after 6 h of aPHF6 stimulation. Similarly, *IL18* and *TLR4* have shown a 20- and 30-fold increase, respectively. After 24 h, however, the inflammation marker gene expressions returned to the basal levels.

### 3.5. aPHF6 Upregulated Caspase-1 Activity, Inflammatory Cytokines, and M2 Markers

Because NLRP3 inflammasome complex upregulation could lead to the secretion of cytokines to make neighboring cells aware of the inflammation, besides the expression of *IL1B,* the CASP1 activity was also analyzed. After 6 h, a significant increase in the CASP1 activity was measured in the cells only after aPHF6 stimulation (Figure 5a). The unaggregated PHF6 showed no effect on CASP1. In the MCC950 pre-incubated cells, no increasedCASP1 activity was measured (Figure 5b), indicating a strong NLRP3 participation in CASP1-mediated cytokine maturation. After 24 h of aPHF6 stimulation, a CASP1 activity increase was absent and not further modulated by MCC950 pre-incubation (Figure 5c,d). Regarding the *IL1B* gene expression, a six-fold increase was observed in 20 μM aPHF6 stimulated cells compared to the PHF6-stimulated and untreated cells after 6 h of stimulation (Figure 6a). However, after 24 h, no changes in the *IL1B* gene expression were detected in either of the treatments (Figure 6d). Western blot observations for the protein expression of the mature IL1β form (17 kDa) revealed a four-fold increase in the expression after 6 h of aPHF6 stimulation, whereas that after 24 h remained comparable to the control group (Figure 6c,f). Note that the IL1β pro-form (31 kDa) was also increased after 6 h but decreased after 24 h in comparison to the control groups (no quantification). With the 20 μM PHF6, however, no significant increase in gene or protein expression was observed at both time points. Moreover, using sandwich ELISA (Figure A6), no color change was observed in comparison to the used positive controls (Nigericin or LPS), indicating no release of IL1β into the supernatant. Then the gene expression of M2 markers such as *IL10*, *MRC1*, *CD163*, and *CD200R1* was measured. Compared to the control, *IL10* and *MRC1* registered an almost 300 times increase in the gene expression after 6 h of aPHF6 stimulation. Similarly, *CD163* and *CD200R1* reported 500-, and 800-fold increases, respectively. After 24 h, however, the marker gene expressions returned to the basal levels (Figure A5). A substantial increase in gene expression of M2 markers was observed compared to M1 markers after 6 h of aPHF6 stimulation.

### 3.6. aPHF6 Induced Autophagy after 6 h in HMC3 Cells

After 6 h of stimulation with aPHF6, the autophagy gene marker Beclin-1 (*BECN1*) was observed to be significantly increased, whereas, after 24 h of stimulation, no alteration in any group was seen (Figure 7a,d). Similar results were measured for Beclin-1 protein levels, which were only significantly elevated after 6 h aPHF6 stimulation (Figure 8b,h). Measurement of additional autophagy markers, such as p62 and LC3I/II, underlined the activation after 6 h (Figure 8d–f). Note that, in contrast to LC3I, the protein levels of LC3II were unchanged at this time point (Figure 8f). No changes were detected after 24 h of aPHF6 stimulation (Figure 8k–m). Further, to verify the autophagy effect, CYTO-ID^®^ 2.0 based autophagic vesicle detection assay was performed, which depicted an almost 1.5 times increase in fluorescence (autophagy vesicle induction) after 6 h of aPHF6 stimulation, as quantified by ImageJ compared to the control group (Figure 7b,c). After 24 h of aPHF6 stimulation, however, the autophagy vesicle induction was comparable to the control (Figure 7e,f). The autophagy induction with the 20 μM PHF6 remained insignificant at both time points.

## 4. Discussion

The failure of Aβ-targeted therapeutic approaches has led to a shift of the focus on tau-targeted strategies [48] and, more specifically, on PHF6 in recent years [49]. A growing body of literature suggests the potential role of microglia in the spread of tau throughout the brain [24,25]. Considering the role of full-length tau in NLRP3 inflammasome activation [27], we hypothesize that the aggregated PHF6, as in the full-length tau, affects the NLRP3 inflammasome causing the overexpression of components triggering the release of IL1β. Our results demonstrate that aPHF6 causes an upregulation of NLRP3 and associated components (ASC, pro-, and matured IL1β) at the gene and protein level. However, using sandwich ELISA, the release of IL1β in the supernatant was not observed, which might indicate that perhaps IL1β was metabolized and therefore was not measurable anymore. The secretion of the mature IL1β form is dependent on non-specific plasma membrane permeabilization followed by pyroptosis [50]. However, no significant membrane integrity loss and LDH release was observed with aPHF6. Limitation of conventional biochemical assays, along with the short-life of matured IL1β, has also been reported [51]. It could finally imply that, though inflammasome is induced, it is not a fully-fledged activated response for the release of proinflammatory cytokines. Generally, NLRP3 activation has been shown to trigger the IL1β release in BV2 mouse microglial cells [52,53]. Moreover, the small size of the hexapeptide could also play a vital role in diminished inflammasome activation.

HMC3 cells used in this experiment were human embryonic cells, derived from 8–10 week old embryos [38]. Ready availability and ease of harvest have established this cell line as a stable experimental human glial model. However, these cells are relatively young (immature) compared to the glial cells of neurodegenerative disease patients and might be devoid of essential cellular pathways that can only be found in adult glial cells. This might also be a possible explanation for the short-term effect of aPHF6, and with a relatively long incubation time, an adaptation of cells brings down the expression to the basal levels. Nonetheless, and as already discussed, based on the signals, microglia can be classified in a simple way as classically (M1) or alternatively active (M2), producing different cytokines, ultimately helping or abating neuroinflammation [54,55]. Our results indicate a substantial increase in the mRNA levels of M2 markers compared to that of M1 state markers. Because M2 is generally regarded as the neuroprotective state of microglia, it can be assumed that microglia perceive the PHF6 stimulation as a mild threat and try to mitigate it by expressing neuroprotective markers/cytokines over time (6 h) and in longer duration (24 h); with an adaptation of the cells, the genetic expression of inflammasome components returns to the basal state. At the early stage of disease onset, M2 microglia predominates in an attempt to reduce inflammation, enhancing homeostasis in relation to neurons [56].

Neuroinflammation and autophagy are closely related to neurodegeneration and protein quality control [57]. An increase in the autophagy limits inflammasome activity by initiating pro-IL1β degradation helping the cells to return to their basal non-reactive state [58]. Further, impaired autophagy polarizes the microglia towards the M1 state; vice versa, restoring autophagy might polarize microglia towards the M2 state [59]. Induction of autophagy could initiate pro-IL1β degradation through autolysosome formation, thereby stopping IL1β release and further inflammation in the cell [60]. The autophagy process was also found to be exacerbated with aPHF6 in our experiments. An increase in autophagy could be related to microglial polarization towards the M2 state. Beclin-1, which is involved in pre-phagophore formation, was found to be overexpressed with 6 h of aPHF6 induction in both mRNA and immunoblot-based studies. However, levels of autophagosome marker LC3-II (a lipidated form of LC3-I) were found unchanged across the samples. This could indicate that, though pre-phagophore formation has been initiated later, autophagy processes were compromised, leading to the non-formation of autophagosome and a failed lysosome fusion. The cytosolic LC3-I, along with p62, was found to be overexpressed in our immunoblot-based studies, indicating a partial impairment in the autophagy flux or complete abrogation [61,62]. The p62 protein involved in protein quality control, as well as autophagy, might accumulate during autophagy deficiency in a cell-type-dependent manner [63]. As described before, the cell line we were using was created from very young cells. An attenuation of autophagy-related proteins—which might be connected to autophagy dysfunction in AD—occurs naturally in older people. Additionally, the peptide we used in this study is artificial and though it is an essential part of full-length tau, does apparently not provide all the same effects as the whole molecule does. Regarding inflammasome and autophagy regulation, additional amino acids seem to be of need. Further, to combat the adverse effect of aPHF6 at 6 h and to stop further inflammation, microglia initiate the autophagy flux for probable degradation of aPHF6, which might negatively regulate the M1 marker or inflammasome expression. Using MCC950, NLRP3 inflammasome involvement has been depicted in elevated CASP1 activity with aPHF6 in our studies. However, the extent of mRNA expression for NLRP3 inflammasome components was also relatively low compared to that of M2 markers, which might indicate that microglia are active and overexpressing NLRP3 to a certain extent; such activation could be counterbalanced by active autophagy and the neuroprotective alternatively active M2 state. Further investigation using co-culture experiments involving microglia and neurons could provide insights into the time-dependent effect of PHF6-stimulated microglial cells on neuronal differentiation and viability.

## 5. Conclusions

Our results demonstrate a time-dependent upregulation of NLRP3, ASC, IL1β, and the autophagy markers Beclin-1 and p62 in human microglial cells in response to the Alzheimer’s PHF6 peptide. Based on the missing release of IL1β into the supernatant and final increase in the autophagosome protein LC3II, we suggest a “priming” of the cells by aPHF6 treatment, but somehow a signal is missing for the final activation of IL1β release and autophagosome formation. Therefore, we conclude that aPHF6 is capable of “priming” the NLRP3 cascade as well as pre-formation of the autophagy process. It might be that other parts of the full-length tau protein and its aggregation are additionally needed for activation for the aforementioned processes. Results from this PHF6 model system-based study might provide further insights into the reaction of human microglial cells to tau-PHF6 stimulation and make microglia attractive for therapeutic modulations. Experiments in the future using full-length tau will give further insights into the activation of microglia cells in the tauopathy in Alzheimer’s disease.

## Figures and Tables

**Figure 1 cells-10-01652-f001:**
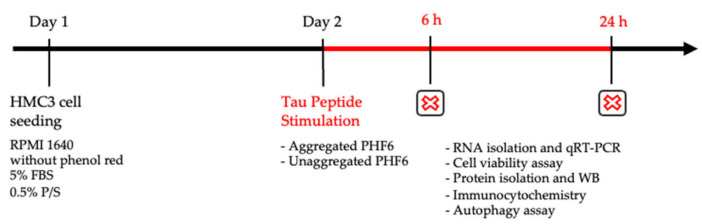
Experimental timeline.

**Figure 2 cells-10-01652-f002:**
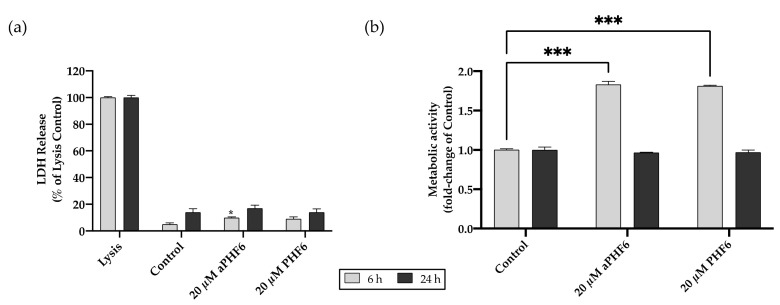
Effect of PHF6 stimulation on HMC3 cell viability and metabolic activity at 6 and 24 h. (**a**) Cell viability assay: Treatment with 20 μM aggregated PHF6 (aPHF6) or 20 μM unaggregated PHF6 (PHF6) did not significantly affect the LDH release at 6 or 24 h. (**b**) Metabolic activity assay: The CTB assay revealed increased metabolism after 20 μM aPHF6 and PHF6 stimulation at 6 h, whereas no changes in any treatment group after 24 h were detected. Maximal lysed cells are treated as 100% or 1.0 positive control. * *p* < 0.05; *** *p* < 0.001.

**Figure 3 cells-10-01652-f003:**
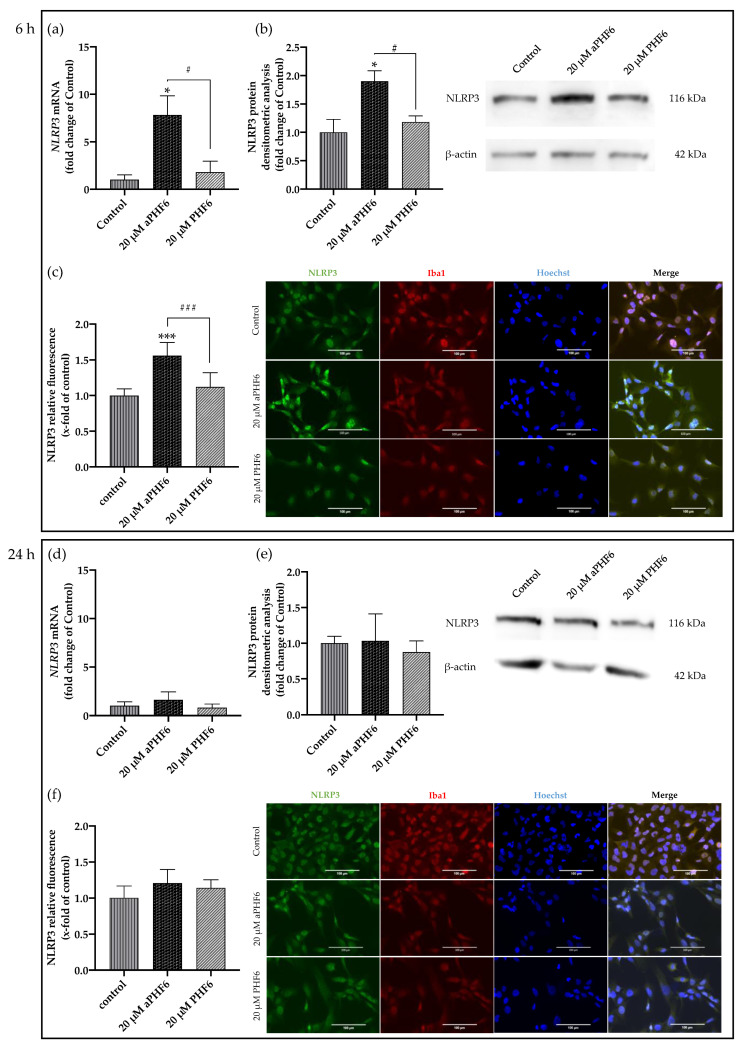
Aggregated PHF6 upregulated NLRP3 mRNA and protein in human microglial cells after 6 h (**a**–**c**), but the effect is absent after 24 h (**d**–**f**). (**a**) Treatment with 20 μM aPHF6 resulted in a seven-fold increase in the *NLRP3* mRNA compared to PHF6 or untreated controls as quantified by qRT-PCR. (**b**) NLRP3 Western blots densitometry revealed a 1.5-fold increase in the protein levels of treated cells compared to control after 6 h of stimulation. (**c**) aPHF6 treatment depicted an increased NLRP3 fluorescence intensity (1.5 times) compared to other treatment groups. (**d**) Gene expression analysis revealed no effect of the treatment with 20 μM aPHF6 on the *NLRP3* expression compared to PHF6 or untreated controls. (**e**) Densitometric analysis of NLRP3 Western blots confirmed no significant changes in protein levels at 24 h. (**f**) However, in some cells an increased NLRP3 fluorescence intensity is observed after the aPHF6 treatment. */# *p* < 0.05; ***/### *p* < 0.001.

**Figure 4 cells-10-01652-f004:**
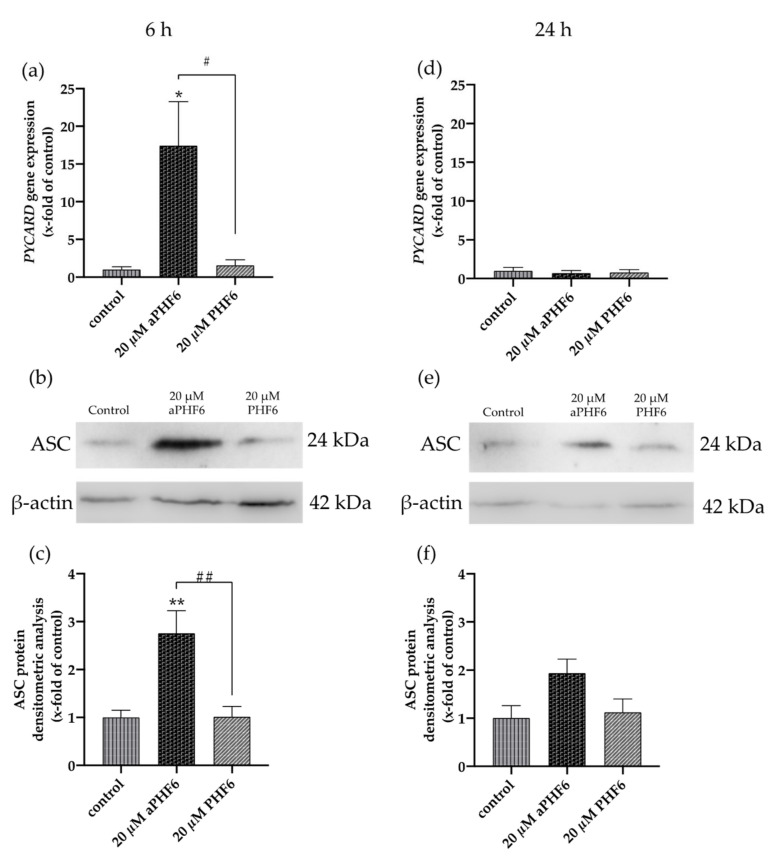
Aggregated PHF6 induced upregulation in *PYCARD* mRNA and ASC protein in HMC3 cells after 6 h. (**a**) An increase in the mRNA levels of *PYCARD* (*ASC*) is observed after treatment with 20 μM aPHF6 in comparison to the PHF6 or untreated controls as quantified by qRT-PCR. (**b**,**c**) Densitometric analysis of Western blots confirmed the increased ASC protein levels at 6 h. (**d**) No changes in *PYCARD* (*ASC*) mRNA levels among different groups were detected after 24 h. (**e**,**f**) However, Western blot analysis reflects a non-significant increase in ASC protein levels in the aPHF6 group compared to the controls. */# *p* < 0.05, **/## *p* < 0.01.

**Figure 5 cells-10-01652-f005:**
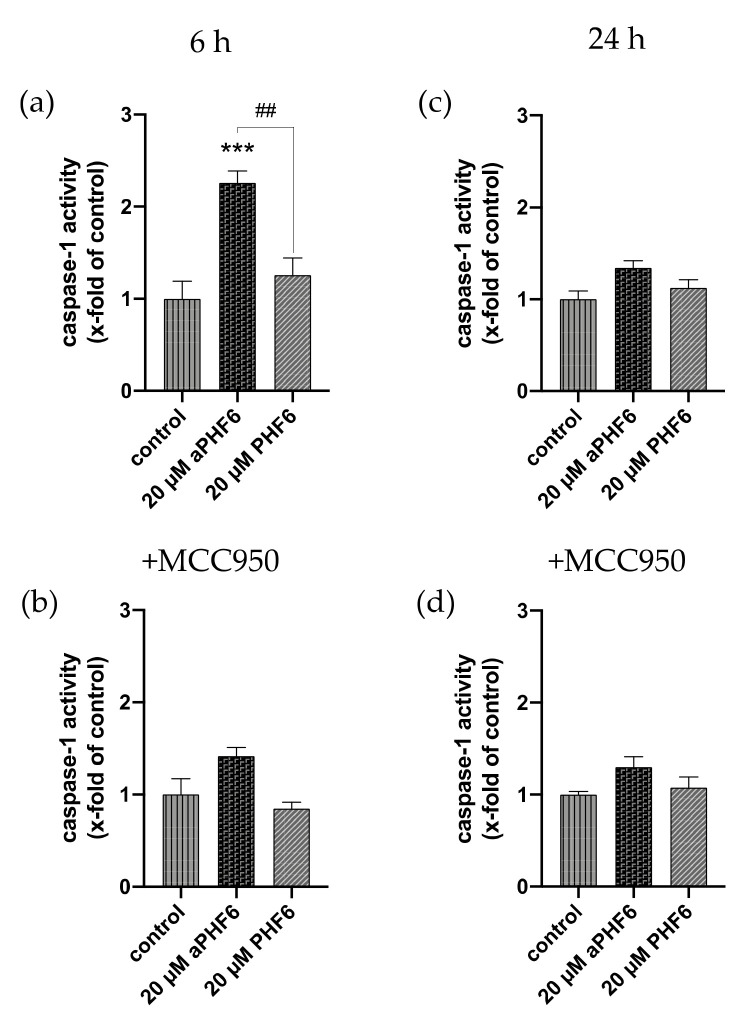
Elevated caspase-1 activity is mediated by NLRP3 after 6 h. (**a**) The determination of caspase-1 (CASP1) activity showcased an increase after 6 h of aPHF6 treatment, which returned to basal levels after 24 h (**b**). (**c**) Using the specific NLRP3 inhibitor, MCC950 revealed that the CASP1 activity was mainly driven by NLRP3 after 6 h. (**d**) At the prolonged stage, no differences were found in comparison to the control and the treatment without the inhibitor. ## *p* < 0.01, *** *p* < 0.001.

**Figure 6 cells-10-01652-f006:**
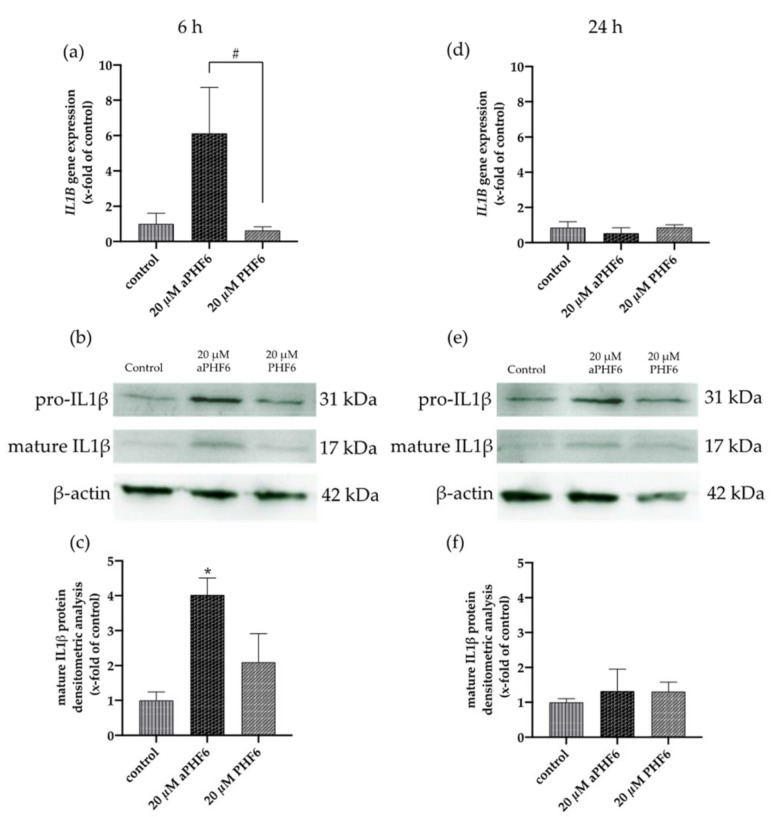
Aggregated PHF6 induced upregulation of IL1β mRNA and protein in HMC3 cells after 6 h. (**a**) An increase in the mRNA levels of *IL1B* was observed after 20 μM aPHF6 treatment in comparison to PHF6 treatment or untreated controls as quantified by qRT-PCR. (**b**,**c**) Densitometric analysis of Western blots confirmed the increased IL1β protein levels in HMC3 cells at 6 h. (**d**) No changes in *IL1B* mRNA levels among different groups were detected after 24 h. (**e**,**f**). Western blot analysis reflected similar IL1β protein levels in all groups. */# *p* < 0.05.

**Figure 7 cells-10-01652-f007:**
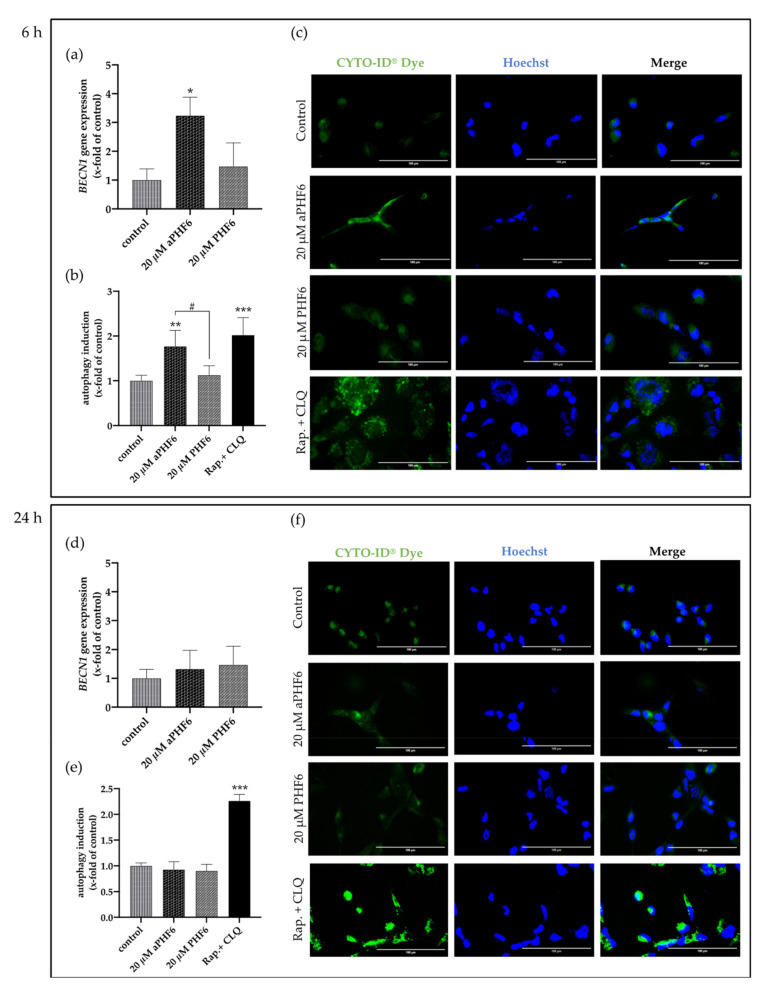
Aggregated PHF6 induced autophagy in HMC3 cells after 6 h. (**a**) Significantly increased mRNA levels of autophagy marker *BECN1* were observed within the 20 μM aPHF6 treatment group compared to PHF6 or untreated controls as quantified by qRT-PCR. (**b**,**c**) The autophagy assay confirmed the increased autophagosome formation after aPHF6 treatment by both plate-reader readout and fluorescence image intensity measurement. (**d**) No changes in *BECN1* gene expression were observed after 24 h of 20 μM aPHF6 treatment. (**e**,**f**) The autophagy assay confirmed no difference in autophagy flux among the treatment groups. A mixture of rapamycin and chloroquine was used as a positive control. */# *p* < 0.05; ** *p* < 0.01; *** *p* < 0.001.

**Figure 8 cells-10-01652-f008:**
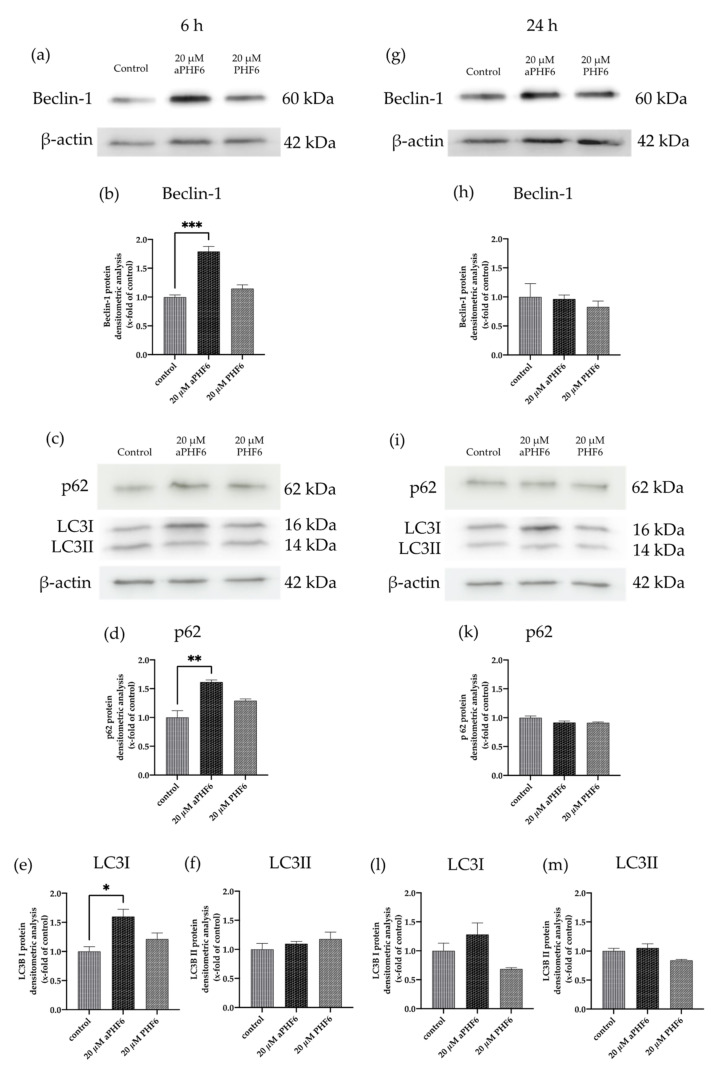
Autophagy markers p62, LC3I, and Beclin-1 are increased after 6 h aPHF6. (**a**,**b**) WB analysis of Beclin-1 confirmed the elevated gene expression after 6 h of aPHF6 stimulation (**g**,**h**), whereas no changes were observed at the later stage. (**c**,**d**) After the short-term stimulation, higher levels of the p62 autophagosome protein were measured, which returned to control levels at the later stage (**i**,**k**). (**e**) On the contrary, LC3I was increased after 6 h, but no changes were detected on the LC3II form (**f**). (**l**,**m**) After 24 h, no significant changes were measured at all. * *p* < 0.05; ** *p* < 0.01; *** *p* < 0.001.

**Table 1 cells-10-01652-t001:** List of primers used for qRT-PCR with their optimum annealing temperature.

Primer	Sequence	AT ^1^ [°C]
*PYCARD*	for: TGGATGCTCTGTACGGGAAG	60
(ASC)	rev: CCAGGCTGGTTGAAACTGAA	
*B2M*	for: TGGGATCGAGACATGTAAGCAG	62
	rev: AGCAAGCAAGCAGAATTTGGAA	
*BECN1*	for: ACGTGGAAAAGAACCGCAAG	60
(Beclin-1)	rev: ACTGAGCTTCCTCCTGATCCA	
*CD163*	for: AAAGTCCAGGAGGAGTGGGG	65
	rev: TCCAAATGCGTCCAGAACCT	
*CD200R1*	for: CTTCATGGCCTGTAAAGATGGC	60
	rev: GGCTGGCCTCTAGGATTAT	
*HPRT1*	for: CCTGGCGTVGTGATTAGTA	60
	rev: AGACGTTCAGTCCTGTCCAT	
*IL1B*	for: CTTCGAGGCACAAGGGACAA	65
(IL1β)	rev: TTCACTGGCGAGCTCAGGTA	
*IL10*	for: GGCATCTACAAAGCCATGAGTG	60
	rev: ATAGAGTCGCCACCCTGATGT	
*IL12A*	for: GCTTTCAGAATTCGGGGAGTG	60
	rev: GTTTGGAGGGACCTCGCTTT	
*IL18*	for: ATGCACCCCGGACCATATTT	64
	rev: TGTTCTCACAGGAGAGAGTTGA	
*MRC1*	for: TGACGAATTGTGGATCGGCT	62
	rev: CGTTACAGGGGTCCCATCAC	
*NLRP1*	for: GAGACAGCTGCCTGACACAT	60
	rev: ACTCCTGGCTTGGAGACTCA	
*NLRP3*	for: CATCGGGGTCAAACAGCAAC	60
	rev: ACGACTGCGTCTCATCAAGG	
*TLR4*	for: TGGATCAAGGACCAGAGGCA	65
	rev: GAGGACCGACACACCAATGA	

^1^ annealing temperature.

**Table 2 cells-10-01652-t002:** List of antibodies used for Western blot.

Antibody	Company	Order-No.	Host	Dilution
β-actin	Santa Cruz, Dallas, TX, USA	sc-47778	mouse	1:5000
ASC	Santa Cruz, USA	sc-271054	mouse	1:1000
*BECN1*	Cell Signaling, Danvers, MA, USA	3495	rabbit	1:1000
LC3II	Thermo Fisher Scientific, USA	PA1-46286	rabbit	1:1000
IL1β	Abcam, Cambridge, UK	ab9722	rabbit	1:1000
NLRP3	Thermo Fisher Scientific, USA	PA5-79740	rabbit	1:1000
SQSTM1/p62	Cell Signaling, USA	5114	rabbit	1:1000
anti-mouse	Sigma–Aldrich, Germany	A4416	goat	1:4000
anti-rabbit	Bio-Rad, USA	170-6515	goat	1:5000

**Table 3 cells-10-01652-t003:** List of antibodies used for immunofluorescence staining.

Antibody	Company	Order-No.	Host	Dilution
ASC	Santa Cruz, USA	sc-271054	mouse	1:500
AIF1 (IBA1)	Millipore, Burlington, MA, USA	MABN92	mouse	1:100
NLRP3	Bioss, Woburn, MA, USA	bs-10021R	rabbit	1:500
anti-mouse 594	Thermo Fisher Scientific, USA	A21203	donkey	1:500
anti-rabbit 488	Thermo Fisher Scientific, USA	A21206	donkey	1:500

## Data Availability

The primary data can be requested by mail from the corresponding author.

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
