# Peer review of "Aggregated Tau-PHF6 (VQIVYK) Potentiates NLRP3 Inflammasome Expression and Autophagy in Human Microglial Cells"

_cells, 2021, doi:10.3390/cells10071652_

Round 1

Reviewer 1 Report

Dear Editor,

The manuscript submitted by Dr. Panda and collaborators entitled “Aggregated tau-PHF6 (VQIVYK) potentiates NLRP3 inflammasome expression and autophagy in human microglial cells” aims at investigating the effects of unaggregated and aggregated forms of the tau-derived PHF6 peptide (VQIVYK) in a human microglia cell line with the purpose of unveiling whether this could alter the expression and activity of the NLRP3 inflammasome and possibly, activate autophagy in these cells.

The authors conduct a whole series of experiments in an immortalised microglial cell line (HCM3) including mRNA, protein and localisation studies (IHC) to test their working hypothesis.

Overall, the study is nicely designed and provides interesting data on PHF6 pathogenic mechanisms in vitro, which can help to better understand the subtle activity of pro-aggregating peptides in Alzheimer’s disease development. There are, however, there are some minor issues that need to be addressed by the authors.

  1. Figure 3 (panels C and F) Please indicate whether relative fluorescence is referred to NLRP3 and possibly also report the levels for Iba1, to which there is no mention in the captions.
  2. Line 360 – “…genetic expression registered a fifteen-fold increase”. Please rephrase as “….gene expression increased by fifteen fold”.
  3. Figure 6 – Measuring cytokine expression using Western blot may prove to be less reliable than measuring the amount of cytokine released in the culture media (ELISA-based assay). Could the authors consider running a multiplex ELISA to measure the levels of cytokines to complement these findings?
  4. I recommend the authors to add the n values to indicate the number of biological replicates used in each experiment. Additionally, it may be worthwhile also indicating how many times experiments were run.
  5. Figure 8 – LC3I seems to be equally increased at 6h and 24h but it is only reported to be statistically significant at the former time point. Can the authors explain the discrepancy?

Author Response

Please find our response letter in the attachment. 

Reviewer 2 Report

In the present study, Panda et al., propose to evaluate the role of tau-PHF6 peptide as a model system to investigate the role of aggregation on microglial polarization, inflammasome activation, and autophagy in the human microglial cell line HMC3.

The major concern with the study is whether it really represents a real contribution in terms of novelty. Although the manuscript has some valuable data, it is already well known the role of peptide fragment PHF6 derived from Tau as a modulator of Tau aggregation, a major component of the plaques of the amyloid β peptide, a hallmark of Alzheimer’s disease (AD). Plus, it is also well described that Tau pathology triggers leads to reactive gliosis (microglia and astrocytes), that combined to high levels of pro-inflammatory molecules (such as inflammasome components) lead to a chronic neuroinflammatory state.

Thus, the whole study was conducted in a microglia cell line without a positive control (using LPS or TNF-alpha to establish the basal cell reactivity). There is no way to make sure that cytokine fluctuations or reactive microglia response are due to the experimental challenge or just simply by the manipulation itself, which compromises the impact of the study and the potential advantages for publishing it.

The minor points are:

  1. Figure 1 can be combined with Figure 2;
  2. Too many figures; It can be combined, making the reading more fluid;
  3. Describe better how the immunofluorescence images were determined using ImageJ software

Author Response

(The authors gave the same response as above.)

Reviewer 3 Report

Aggregation of tau protein to toxic neurofibrillary tangles is a key feature of Alzheimer’s Disease (AD), the leading cause of dementia worldwide. Inflammation is activated at early stages of  the AD onset/development. In the present manuscript, the authors investigated the effects of aggregated and unaggregated forms of the tau-derived PHF6 peptide(VQIVYK) on HMC3, a human microglial cell line .

They report that aggregated VQIVYK  increases the NLRP3- associated components (ASC, pro- and matured IL1b) at both mRNA and protein levels in a dose- and time-dependent manner leading increased expression(but not secretion) of IL1β in HMC3 cells. Fibrillar PHF6-peptide also activates other related inflammation markers and microglial polarization (M1 state towards M2 state). Furthermore, they show a time-dependent effect of the aggregated PHF6 on BECN1 expression whereas only the cytosolic  LC3I ( but not lipidated LC3II) and autophagic p62 are elevated. As authors reported in the discussion and conclusion, aggregated PHF6 treatment appears to cause only a  “priming” effect of the cells on the NLRP3 cascade. The manuscript is well-written but additional experiments are needed  before the paper is accepted for publication.

Major revisions

NLRP3–ASC inflammasome becomes activated by a dual stimuli leading to heteromer formation of ASC and activation of caspase-1. Indeed In most cell types, NLRP3 must be “primed”, and a prototypical  example of such a priming event is the binding of LPS to TLR4. Once primed, NLRP3 can respond to its stimuli and assemble the NLRP3 inflammasome (Guo et al., 2015). Aggregated tau activates is reported to activate the LRP3-ASC inflammosome in LPS-stimulated primary microglia (Stancu et al., 2019). The referee suggests that authors should perform all the experiments in HMC3 cell line following treatment with LPS, LPS+Nigericin, LPS+aggregated PHF6. A comparison with results shown in the present manuscript ( in the absence of LPS stimulation) will be helpful to provide a more complete picture of the cascade and to draw conclusions.

Author Response

(The authors gave the same response as above.)

Round 2

Reviewer 2 Report

The authors have addressed satisfactorily the points I raised previously. With the additional data presented and discussion, the paper has improved.

Author Response

Thank you. We appreciate your comments and suggestions.

Reviewer 3 Report

The authors should include the data concerning LPS and nigericin (used as positive control) as supplemental material in the revised version of the manuscript 

Author Response

Thank you for the suggestion. We have added the figure in the appendix as A6 and referred in the results section to it.